# Gradual Response of Cyanobacterial Thylakoids to Acute High-Light Stress—Importance of Carotenoid Accumulation

**DOI:** 10.3390/cells10081916

**Published:** 2021-07-28

**Authors:** Myriam Canonico, Grzegorz Konert, Aurélie Crepin, Barbora Šedivá, Radek Kaňa

**Affiliations:** 1Centre Algatech, Institute of Microbiology of the Czech Academy of Sciences, Opatovický Mlýn, 379 81 Třeboň, Czech Republic; canonico@alga.cz (M.C.); konert@alga.cz (G.K.); crepin@alga.cz (A.C.); sediva@alga.cz (B.Š.); 2Faculty of Science, University of South Bohemia in České Budějovice, Branišovská 31a, 370 05 České Budějovice, Czech Republic

**Keywords:** high light, thylakoid membrane, microdomains, carotenoids, photoprotection, *Synechocystis*, non-photochemical quenching, photoinhibition, photosystems

## Abstract

Light plays an essential role in photosynthesis; however, its excess can cause damage to cellular components. Photosynthetic organisms thus developed a set of photoprotective mechanisms (e.g., non-photochemical quenching, photoinhibition) that can be studied by a classic biochemical and biophysical methods in cell suspension. Here, we combined these bulk methods with single-cell identification of microdomains in thylakoid membrane during high-light (HL) stress. We used *Synechocystis* sp. PCC 6803 cells with YFP tagged photosystem I. The single-cell data pointed to a three-phase response of cells to acute HL stress. We defined: (1) fast response phase (0–30 min), (2) intermediate phase (30–120 min), and (3) slow acclimation phase (120–360 min). During the first phase, cyanobacterial cells activated photoprotective mechanisms such as photoinhibition and non-photochemical quenching. Later on (during the second phase), we temporarily observed functional decoupling of phycobilisomes and sustained monomerization of photosystem II dimer. Simultaneously, cells also initiated accumulation of carotenoids, especially ɣ–carotene, the main precursor of all carotenoids. In the last phase, in addition to ɣ-carotene, we also observed accumulation of myxoxanthophyll and more even spatial distribution of photosystems and phycobilisomes between microdomains. We suggest that the overall carotenoid increase during HL stress could be involved either in the direct photoprotection (e.g., in ROS scavenging) and/or could play an additional role in maintaining optimal distribution of photosystems in thylakoid membrane to attain efficient photoprotection.

## 1. Introduction

Photosynthesis is a key, light-driven bioenergetics process on Earth. The light-dependent photosynthetic reactions take place in the thylakoid membrane (TM). The membrane is located either in chloroplast of eukaryotic plants and algae or in the cytoplasm of prokaryotic cyanobacteria. Light-photosynthetic reactions in TM include various processes from light-absorption to electron/proton transport. For the purpose of light harvesting in cyanobacteria, several large and highly pigmented protein complexes localized in the TM are employed, including photosystem I (PSI), photosystem II (PSII), and phycobilisomes (PBS; the light-harvesting antennae bound on the TM surface). PSI, PSII, and PBS can then be for simplicity’s sake called pigment–protein complexes (PPCs) [1].

The mechanisms behind the PPCs’ activity (light absorption, charge separation, electron transport, etc.) have already been described in vitro for all these isolated complexes [2]. However, much less is known about overall cooperation and interactions between the PPCs in native TMs and how it is interlinked with their heterogeneous localization in vivo [1]. The meaning of the spatial PPCs’ heterogeneity is an open question in photosynthesis [1,3,4,5,6]. It has been clearly shown almost 40 years ago [7,8,9] that, in higher plants, PSII is mainly present in TM regions called grana (stacked TM areas), while PSI is found in stroma lamellae (unstacked TM areas). A similar grana/stroma-like heterogeneity has been recently identified also in cyanobacteria [1], in line with several other works showing spatial variability in PPC localization in thylakoids [10,11,12,13,14]. In the work of Strašková and co-workers [1], a functional analogy between granal/stromal thylakoid of higher plants TM and so-called photosynthetic microdomains (MDs) in cyanobacteria has been proposed. Those MDs represent specific clusters of TM co-localization with a characteristic PSI/PSII/PBS ratio. Importantly, particular MDs do not fully segregate PSI/PSII and PBS from each other [1]. The importance of the MD mosaic or PPCs’ heterogeneity in TM for cyanobacterial photosynthesis has been intensively studied [1,3,4,5,6]. However, its variability in PPCs’ co-localization caused by fluctuation in light is still poorly understood (see some recent results [3,5,6,15,16,17]).

Generally, MDs’ organization is rather stable in the range of minutes [1], but it can vary in the range of hours or days [6]. This slow process of TM acclimation of *Synechocystis* sp. PCC 6803 can be triggered at variable growth light conditions (continuous light vs. light/dark cycle) and it resembles higher plants grana/stroma reorganization defined as TM plasticity [6]. However, the stability/flexibility of TM organization remains an open question. Some authors suggested rather dynamic behavior of PPCs at extremely high-light (HL) irradiation [3,15] or static behavior of PPCs in a minute’s time scale at more physiological HL intensities [1,5]. Therefore, the effect of HL stress on the overall TM and PPCs’ organization is still not fully clear.

Light plays an essential role in photosynthesis but can be harmful in the case of excess. Therefore, all phototrophs including cyanobacterial cells can trigger several fast (in the range of minutes) photoprotective mechanisms that are well defined based on bulk measurements (i.e., on cell population level [18]). The three most important strategies are represented by photoinhibition [19], phycobilisome decoupling [20] and non-photochemical quenching (NPQ) [21]. Photoinhibition, in its broader and the original sense, represents a light-induced decrease in the photosynthetic rate or photosynthetic efficiency measured either as the oxygen evolution or CO_2_ assimilation rates [22]. On the molecular level, photoinhibition is connected with PSII degradation, because PSII complexes are more sensitive to HL than PSI [23]. The decrease in PSII activity is caused by degradation of the 32-KDa D1 protein [24,25,26,27,28,29,30] that needs to be newly synthesized and assembled into PSII complex [31] to recover PSII activity. Therefore cyanobacteria, under photoinhibition, display a decrease in the levels of active PSII that can be measured indirectly as a change in variable chlorophyll (Chl) a fluorescence [32] or directly by the rate of D1 protein turnover [33,34].

The second photoprotective mechanism found in cyanobacteria is represented by PBS decoupling (reviewed in detail in [20]). In a situation of stress condition, such as HL or non-optimal (low or high) temperature, PBS can be partially decoupled [16,35], either functionally or physically detached from the reaction centers [16,17,35,36]. It has been proposed that the uncoupling process could be followed either by PBS redistribution between the two photosystems or disassembly of longer-term detached PBS from the TM [20]. Interestingly, this mechanism has been identified also during the long-term (in days) acclimation of thylakoid membrane to various cultivation light conditions [6]. The third photoprotective mechanism, NPQ, is also detectable based on changes in variable Chl a fluorescence [37]. Even though some previous works have discussed cyanobacterial NPQ as a PSII reaction center type quenching [38], the dominant part is represented by fluorescence quenching in PBS [39]. The mechanism is triggered by blue light and represents a unique cyanobacterial strategy to dissipate excessive light absorbed by PBS. The process requires protein called orange carotenoid protein (OCP) that acts as a blue light sensor [40]. The protein contains a carotenoid, echinenone that acts as a sensing molecule. Interestingly, carotenoids have an essential role in photoprotection in general, as shown for many other phototrophs [41,42,43]. Generally, carotenoids can be involved in the quenching of chlorophyll triplet state [44], in the scavenging of ROS [45] and changes in membrane properties [46], that can affect the efficiency of light harvesting [47,48]. However, the role of carotenoids in cyanobacterial photoprotection (except for the OCP-based mechanism) is less understood [21,49]. Cyanobacterial cells, indeed, contain a large amount of myxoxanthophyll, zeaxanthin and other carotenoids [50,51]. In contrast to higher plants, in which these pigments seem to act mainly as part of the PPCs, several studies pointed to their presence and importance in cyanobacteria in both TM and plasma membrane [38,40,41].

In the present paper, we studied the overall response of cyanobacterial thylakoids to acute HL stress. We identified a multiphase response of cyanobacteria to the HL stress (in line with [5]), with three main phases: (1) fast response phase (0–30 min), (2) intermediate phase (30–120 min), and (3) slow acclimation phase (120–360 min). During these three phases, cyanobacterial photosynthesis was affected on several levels including: (1) protein/pigment composition, (2) TM microdomain organization, and (3) activation of photoprotective mechanisms. In more details, we observed a more even distribution of PPCs on HL as it was visible in a smaller spatial variability in PSI/PSII/PBS distribution per cell. Simultaneously, we identified fast activation of photoinhibition and non-photochemical quenching (first phase) and temporal process of PBS decoupling (second phase). Finally, HL stress also caused the accumulation of carotenoids, myxoxanthophyll and γ-carotene. We suggest that the carotenoid accumulation during the acute HL stress (6 h) could help to maintain photosynthetic function either by direct photoprotection (e.g., ROS scavenging), or through indirect tuning in TM fluidity.

## 2. Materials and Methods

### 2.1. Growth Conditions and High-Light Treatment

We used the PSI-YFP tagged strain derived from the glucose-tolerant strain of *Synechocystis* sp. PCC 6803 [52] previously described in Tichý et al. [53]. Photosystem I was fluorescently tagged on its PsaF subunit as described previously [1,54] (hereafter *Synechocystis PSI*-*YFP)*. Cells were grown on an orbital shaker (T = 28 °C) in BG11 medium at continuous light (35 μmol photons m^−2^ s^−1^, fluorescent tubes). Cells in the exponential growth phase and with low concentration (OD_730_ = 0.2–0.3) were used for the experiments. OD_730_ was measured with a WPA S800 Diode Array Spectrophotometer (Biochrom Ltd., Cambridge, England). High-light (HL) treatment was performed under white diodes (700 μmol photons m^−2^ s^−1^) at 28 °C during continuous shaking. The control cells were kept at 35 μmol photons m^−2^ s^−1^ at 28 °C during continual shaking for the same time as for the HL treated cells. Cells were used for measurements at time intervals of 0, 10, 30, 60, 90, 120, 240, 360 min. The minimal effect of YFP tagging at PSI on the cell physiology and protein composition has been already proved before [1]; we additionally tested the effect of YFP tagging on light sensitivity of *Synechocystis* PCC 6803 strain. We did not observe significant changes between WT strain (without YFP tag at PSI) and *Synechocystis* PSI-YFP strains considering PSII photochemistry, pigment composition, single-cell fluorescence and others. Therefore, only the PSI-YFP tagged strain of *Synechocystis PSI-YFP* was used to study the HL effect.

### 2.2. Characterization of Cell Sizes, Types and Counts

Cell sizes, counts, and types during HL treatment were monitored by cell counter (Beckman, Multisizer 4, Indianapolis, IN, USA) and by confocal microscope (Zeiss LSM 880; Carl Zeiss Microscopy GmbH, Oberkochen, Germany). Cells were counted by a Coulter counter equipped with 50 μm aperture, where size threshold level was set up in the 1–4 μm range. A total of 50 μL of each sample was diluted in 10 mL of electrolyte solution (0.9% NaCl in deionized water) and measured 3 times. For comparison, the individual cell counts and types were also extracted from the confocal images according to Konert et al. [5] and Canonico et al. [6]. In line with Canonico and co-workers [6], we analyzed 4 characteristic cell shapes (regular, elongated, dividing, or string) separately. Other used parameters include fluorescence intensities and size.

### 2.3. Biochemical Analysis of Pigment–Protein Complexes

Proteins were separated from isolated thylakoid membranes. First, cells were pelleted, washed, and resuspended in buffer B (25 mM MES/NaOH, pH 6.5, 10 mM CaCl_2_, 10 mM MgCl_2_, and 25% (*v*/*v*) glycerol). Then, they were broken using balotina/zirconia beads and Mini-Beadbeater-24 (BioSpec Products, Bartlesville, Oklahoma, USA) for 5 cycles of 30 s of breaking and 1 min in ice. Thylakoid membrane fraction was separated by centrifugation at 20,000 *g* for 30 min at 4 °C. The fraction containing thylakoid membrane proteins (Chl content 5 μg) was loaded in the clear-native polyacrylamide gel electrophoresis (CN-PAGE). Protein complexes from the membrane were solubilized with 10% dodecyl-β-D-maltoside (DDM) in water to obtain the sample volume/DDM = 10 *v*/*w*). Native protein complexes were separated on 4%–14% gradient polyacrylamide gel (acrylamide to bis-acrylamide ratio was 60:1) according to Wittig et al. [55]. Native gels were color-scanned and chlorophyll a fluorescence was obtained by LAS 4000 camera (Fuji, Boston, MA, USA). The color gel pictures were analyzed by the ImageJ software (FIJI distribution). Each band corresponding either to PSI (trimer or monomer) or PSII (dimer or monomer) was taken per each time point. Areas of the peaks were analyzed and normalized to the time 0 min.

### 2.4. Confocal Microscopy

Cells for confocal imaging were prepared and images acquired by a method slightly modified from our previous papers [1,5]. We used a laser scanning confocal microscope (Zeiss LSM 880, Carl Zeiss Microscopy GmbH, Germany) equipped with a plan-apochromatic 63x/1.4 Oil DIC M27 objective. The three channel pictures were obtained by 2 sequential images with different parameters. PBS emission was excited by 633 nm laser (dichroic mirror: MBS 488/543/633) and detected at 642–677 nm (pixel dwell time between 8 and 33 µs). In the following sequence the chlorophyll a autofluorescence from PSII and YFP fluorescence from PSI were both excited with 488 nm Argon laser (dichroic mirror: MBS 488/543/633) and detected at 696–758 nm and 526–580 nm, respectively (pixel dwell time between 8 and 33 µs). Images were acquired with 8 bit, 512 × 512 pix).

Acquired images were separated into individual cell pictures by our ImageJ script (Fiji distribution, ver 1.53c) and their shapes were assigned according to equations described in detail by Canonico et al. [6]. Total cell fluorescence was calculated from whole cell area, and histograms for inner/outer area of cell were obtained by separation of cells into these two regions. Outer cell region was marked from end of cell towards inside of cell in thickness of 10 pixels (520 nm). The remaining core area was marked as inner region.

### 2.5. Measurements of Variable Chlorophyll a Fluorescence

The maximal quantum yield of PSII photochemistry (parameter F_v_/F_m_) was measured by AquaPen-C fluorometer (AP–C100, Photon Systems Instruments, Brno, Czech Republic) with 630 nm excitation and a 667–750 nm detection range. Maximal fluorescence (F_m_) minimal (F_0_) and variable (F_v_ = F_m_ − F_0_) for Chl a fluorescence in dark adapted cells (for 20 min) were used to estimate maximal PSII efficiency F_v_/F_m_.

The extent of non-photochemical quenching (NPQ) was measured by a DUAL- PAM-100 fluorometer (Heinz Walz GmbH, Effeltrich, Germany) with a Dual-DR detector head (excitation 620 nm, detection above 700 nm). Cells were dark adapted for 10 min before the experiment. The values for estimation of NPQ were acquired from a standard protocol for cyanobacteria (see Appendix A). At first, the minimal fluorescence (F_0_) was detected for dark adapted cells, in State II, by a red measuring light (20 Hz, 10 μmol photons m^−2^ s^−1^, λ = 620 nm). Subsequently, cells were transited to State I by low intensity blue light (80 μmol photons μmol photons m^−2^ s^−1^, λ = 460 nm, duration 180 s) and the maximal fluorescence (F_m_) was obtained by a multiple turnover saturating flash (red light λ = 620 nm, 4000 μmol photons m^−2^ s^−1^, duration 400 ms). The NPQ was then induced by high-intensity blue light (1374 μmol photons m^−2^ s^−1^, duration 180 s) and subsequent recovery from NPQ state was then measured in low blue light (80 μmol photons μmol photons m^−2^ s^−1^, λ = 460 nm, duration 360 s). The maximal value of fluorescence for the light adapted sample (F_m_′) was estimated by a multiple turnover flash (red light λ = 620 nm, 4000 μmol photons m^−2^ s^−1^, duration 400 ms) every 10–30 s of the protocol. F_m_′ values were then used to estimate NPQ based on the Stern–Volmer relationship (NPQ = (F_m_–F_m__′_)/F_m′_). The presented steady-state values of NPQ then represent F_m_′ after 180 s on high light and its recovery on low blue light for 360 s.

### 2.6. Fluorescence and Absorption Spectroscopy

Low temperature fluorescence emission spectra were recorded at 77K by an SM-9000 spectrofluorometer (Photon Systems Instruments, Brno, Czech Republic) (see, e.g., [56] for details). Cyanobacterial cells were dark adapted for 20 min and then concentrated on a GF–F filter (Whatman, Maidstone, UK). The filter with cells were then immersed into a Dewar flask with liquid nitrogen. Fluorescence was excited by monochromatic diodes (λ = 450 or 530 nm) and detected by diode array detector (spectral bandwidth 0.8 nm) in the spectral range of 200–980 nm. The dark current of the instrument was automatically subtracted before measurements, based on the method described previously for room temperature measurements [57]. For data processing, the spectra of cells were baseline-corrected using a blank sample (filter with BG-11 media).

Room temperature absorption spectra were measured by a Unicam UV/VIS 500 spectrometer (Thermo Spectronic, Cambridge, UK) equipped with an integrating sphere. Cells were collected on the membrane filters (pore size 0.8 mm; Pragochema, Prague, Czech Republic) and spectrum in the range 350–800 nm was detected.

### 2.7. Characterization of Pigment Composition

Pigment concentrations in cells were estimated spectroscopically (absolute chlorophyll concentrations by Ritchie’s method [58]) and by HPLC analysis (relative pigment content). Pigments were isolated from 1 mL of cell suspension (OD range 0.3–0.4) that was centrifuged (10 min, 2000 g). The supernatant (990 μL) was removed, and the pellet was re-suspended in 990 μL of methanol (100%), vortexed and kept for 20 min on ice in the dark. The prepared sample was then centrifuged (10 min, 2000 g) and the supernatant with pigments in methanol solution was then used for absorption and for HPLC pigment analysis by an Agilent-1260 HPLC system (Perkin Elmer, Boston, MA, USA) equipped with a Radiomatic 150 TR scintillation detector and a diode-array detector.

The extracted pigments were injected into the HPLC system and pigments were separated on a Zorbax Eclipse Plus C18 column (4.6 μm particle size, 3.9 × 100 mm; Agilent, Santa Clara, CA, USA) on a linear gradient of two solvents: A (35% methanol, 15% acetonitrile in 0.25 M pyridine) and B (20% methanol, 20% acetone in acetonitrile). Pigments were eluted with solvent B (30%–95% in 25 min) followed by 95% solvent B at a flow rate of 0.8 mL min^−1^ at 40 °C, as described in [59]. Eluted pigments were identified based on their absorption spectra and retention times. The presence of γ-carotene in cells was further identified by comparison with the HPLC analysis of the commercially purchased standard of ɣ-carotene (analytical standard, purity ≥95%, Sigma Aldrich, Saint Louis, MO, USA). The pigments were identified based on their absorbance at 440 nm. Resulting values were normalized to total pigment absorption at 440 nm for all HPLC peaks. The absorption spectra of extracted pigments from cell suspension were collected by Unicam UV/VIS 500 spectrophotometer (Thermo Spectronic, Cambridge, UK). Absorptions at specific wavelengths (665 nm and 720 nm) were used to calculate chlorophyll a concentration (μg/mL): Chl a = 12.9447 (A_665_−A_720_) [58].

## 3. Results

### 3.1. High-Light Treatment and Cyanobacterial Cell Size

Changes in the size of *Synechocystis PSI-YFP* during 6 h of HL treatment were estimated by cell counting (Figure 1). Data were considered in light of the three-phase behavior published recently [5]. Total number of cells increased only slightly during HL, and there was only a slight difference between HL and the control samples (Figure 1a). The similar increase in the number of cells (for both HL and control) was also visible in the parameter of optical density (OD_730_—Appendix A); however, that reflects both difference in cell sizes and/or cell counts [60]. Therefore, we checked the relative contribution of smaller (diameter ≤1.85 µm) and bigger (diameter >1.85 µm) cells during HL stress directly by cell counter (Figure 1b). Control cell values were stable over time, in contrast to the HL sample with the increased/decreased number of bigger/smaller cells (Figure 1b)—this was visible during the third phase (120–360 min) of HL response. We wanted to observe if we could see the same trend at the single-cell level. Our automatized approach in cell imaging (see Materials and Methods) allowed us to test whether the increase in the count of the bigger cells at HL was due to the higher amount of dividing cells [6]. However, we could not prove/disprove the hypothesis that the cell cycle was affected by HL treatment (Appendix A). Therefore, we did not obtain direct proof at the single-cell level to see if *Synechocystis PSI*-*YFP* cell division was delayed by HL treatments as it is indicated by changes in the cell sizes (Figure 1b).

### 3.2. Changes in Organization of Pigment–Protein Complexes during HL Treatment

Because the HL treatment partially affected the proportion of bigger cells (Figure 1b), we checked whether it resulted in different protein heterogeneity in thylakoids. We estimated the effect of acute HL stress on the pattern of PPC co-localization into MDs (Figure 2), as defined by Strašková et al. (2019) [1]. We showed the existence of heterogeneous membrane areas in thylakoids called MDs based on co-localization of the three main PPCs: PSI (tagged by YFP), PSII, and PBS. Our data were in line with several previous results [1,5,6,61]. The three-channel imaging of these complexes gave rise to a mosaic of microdomains in TM that is visible as RGB (red–green–blue) pictures (see the color scheme in Figure 2a). It shows heterogeneous organization of PSI, PSII, and PBS inside thylakoids that is also confirmed by grey-scale pictures (Figure 2a). Particular MDs in the TM mosaic are then represented by membrane areas of specific PSI/PSII/PBS ratios and they represent specific cluster inside TM. MDs are visible as differently colored areas of TM (see RGB picture in the Figure 2a). In line with previous data [1,5,6,61], the two most dominant areas were represented by magenta MDs (with dominant PSII and PBS content, less PSI) and green MDs (with dominant PSI, less PSII and PBS content), both representing an analogue of granal/stromal areas of plant thylakoids [1].

We studied changes in MDs during HL treatment. We have analyzed distribution of MDs inside single-cell thylakoids (see “RGB” line of Figure 2b,c) and changes in the intensity of their membrane areas for all three PPCs (see heatmap pictures of “PSI-YFP”, “PSII”, and “PBS”). Changes were analyzed separately for regular (reg), elongated (elo), dividing (div), and string (str) shapes of cells representing different stages in the cell cycle (see [6] for details). We observed a general decrease in fluorescence intensity of all three PPCs during HL (Figure 2c); the effect was absent in the control sample (Figure 2b). Further, we also observed a significant decrease in spatial PPCs’ heterogeneity per single cell during HL (see disappearance of red spots in the heatmap of PSI, PSII, and PBS in Figure 2c). We noticed that all PPCs were more evenly distributed as a result of HL stress (compare redundancy of red spots in Figure 2b,c). This was further confirmed by the decrease in the fluorescence signal variability (visible as standard deviation value) per single cell for all three PPCs (data not shown). On the other hand, HL did not significantly affect presence of grana-like (magenta) and stroma-like (green) MDs (see RGB line in Figure 2c). In summary, our data suggest that the HL treatment induced only a loss in the intensity of high-emission spots (see red spots in the heatmap of Figure 2b) because PSI-YFP, PSII, and PBS became more evenly distributed in the thylakoids after acute HL stress.

### 3.3. Changes in the Single-Cell Fluorescence during HL Stress

The observed decrease in the fluorescence intensity visible in pictures (Figure 2) was further quantified and calculated for all three PPCs per single cell from pictures (Figure 3). Cells exposed to HL lost the fluorescence of PBS, PSI, and PSII over time (Figure 3b) in contrast to stable values for control cells (Figure 3a). The relative changes in the PSI/PSII/PBS fluorescence (relative to sum fluorescence values of all 3 PPCs) showed the three distinct phases in HL treated cells (Figure 3d), in line with our previous work [5]. The new set of data represented almost six thousand individual cells from six biological replicates (Figure 3). The first phase (fast response phase, up to 30 min) was characterized by an increase in relative PSI fluorescence ratio. The second phase (intermediate phase; 30–90 min) was characterized by stabilization of relative dynamics of all fluorescence ratios (Figure 3d). The following third phase (slow acclimation phase; 120–360 min) was then characterized by a fluorescence decrease in the relative PSI signal per cell and the concomitant increase in the relative PBS signal (Figure 3d). It is important to note that the relative PSII signal per cell was the most stable from all three PPCs during HL treatment (Figure 3d). In summary, the relative PSI-YFP fluorescence per cell initially increased and simultaneously PBS decreased during the first 30 min. These relative values seem to progressively come back to their initial points during the final slow acclimation phase (120–360 min; Figure 3d). It shows that, although we observed absolute fluorescence decay during HL due to quenching and degradation processes (Figure 3b), the relative ratios of PSI, PSII and PBS (all normalized to sum fluorescence of all three PPCs) per cell tend to recover to almost the same values as the initial non-stressed condition (Figure 3d).

### 3.4. Effect of HL Treatment on Photosystem II Photochemistry

We identified a three-phase acclimation response of *Synechocystis PSI*-*YFP* at the single-cell level. We aimed to address whether the three-phase behavior on the single-cell level (Figure 3) correlates with some functional photoprotective processes known from bulk measurements of variable fluorescence. Therefore, we studied the progress of photoinhibition (measured as decrease in F_v_/F_m_—the maximum quantum yield of PSII, Figure 4a) and NPQ (Figure 4b). Indeed, F_v_/F_m_ decreased significantly during the first phase (up to 30 min) due to HL treatment (Figure 4a). The kinetics of F_v_/F_m_ decay slowed down in the second, intermediate phase (30–120 min), and remained constant in the slow acclimation phase (120–360 min), in line with our previous data [5]. These results point to the photoinhibition-induced increase in the D1-protein turnover [62] that is visible during the first 2 h on HL (Figure 4a).

We measured the NPQ parameter during HL treatment (Figure 4b) and found out that NPQ increased mostly during the first two phases (Figure 4b), when PBS emission per cell was relatively lowered (Figure 3d). Later, in the last slow acclimation phase (120–360 min), NPQ remained constant. Interestingly, when we calculated the extent of NPQ recovery in the dark (NPQ_rec_ calculated as remaining quenching after 802 s from the start of the protocol), we clearly saw that the recovery was even faster at the final (slow acclimation) phase of the HL treatment. It could indicate that the process of OCP reversibility from its active to inactive form was accelerated for long-term HL treatment (see discussion for details). We have recently identified a temporal importance of PBS decoupling during a long-term (weeks) TM reorganization [6]. We studied the same phenomenon considering the three-phase response of single cyanobacterial cells to HL (Figure 4c,d). The calculated F_654_/F_692_ ratio of 77K fluorescence represented the ratio of PBS/PSII emission (Figure 4d). Its increase indicates the presence of an energetically decoupled PBS. This ratio only slightly increased during the first phase (0–30 min) but became significantly unstable and very high during the second phase (30–120 min) (Figure 4d), as it is visible as a change in the full fluorescence emission spectra (Figure 4c). It indicates that PBS decoupling becomes important mostly during the second phase of TM reorganization at HL (30–120 min, see the point at 90 min). We tried to address the process of PBS decoupling on the level of single *Synechocystis PSI*-*YFP* cell (Appendix A). We addressed the basic question of whether energetically decoupled PBS (visible in 77K fluorescence, Figure 4c) remains inside the TM area or if PBS physically move during the process of decoupling (see, e.g., [16,17]); in this case it should be visible as fluorescence increase in the central cell area. For that purpose, we calculated PBS fluorescence emission in these two cell areas (Appendix A), but we found no significant differences between fluorescence in peripheral membrane area and central cell area (see Appendix A). In both cases PBS fluorescence was quenched, as it is visible in the shifting in the histograms from higher to lower fluorescence. The recalculated relative changes in both areas (compared to time 0 min) finally proved that PBS fluorescence was rather quenched in both these areas both at 90 min and 360 min of HL treatment (Appendix A). It indicates that PBS were not physically re-localized from the TM into the central part of cells during our HL treatment (see discussion).

### 3.5. Effect of HL Treatment on PPCs Levels

We analyzed pigment–protein composition during HL treatment by clear native gels (CN) and by 77K fluorescence spectra of intact *Synechocystis PSI*-*YFP* cells (Figure 5). CN gels were scanned as color pictures (Figure 5a) and as Chl autofluorescence (Figure 5b) to estimate the position of photosystems in the gel. The most pronounced change we observed was a clear vanishing of PSII dimer (PSII [2]) that remained less abundant until the end of experiment (Figure 5b). It was also supported by a detailed gel picture analysis by Image J (Appendix A). This is in line with the maximal extent of photoinhibition visible in F_v_/F_m_ (Figure 4a) that was noticeable just before the end of the second phase. Based on the similar quantification, we analyzed PSI bands’ intensity. Here, we were also able to identify even smaller changes in PSI oligomerization state (trimer vs. monomer PSI—see PSI [3]/PSI [1] (Appendix A)). The PSI oligomerization showed a three-phase pattern as PSI [3] initially increased (up to 30 min), and then more PSI [1] appeared later (time 90 min) in the second phase (Appendix A). The oligomerization of PSI, then, did not change and remained stable in the final slow acclimation phase (120–360 min).

We also investigated changes in PSI/PSII ratios based on the 77K fluorescence emission spectra (Figure 5c,d), as the F_724_/F_692_ fluorescence ratio can be used as a proxy for PSI/PSII ratio (see, e.g., [56]). In line with the results observed on gel, the PSI/PSII initially increased during the first phase (0–30 min), then it stabilized during the second phase (30–120 min) and finally progressively decreased and returned to values similar to the initial levels (before starting the HL stress) during the third slow acclimation phase (120–360 min). It shows that the systems of protein complexes (in this case reflected as PSI/PSII ratio, Figure 5d) acclimated to the new conditions (HL stress); similar signs of successful acclimation are also visible on single cell measurements of PSI/PSII/PBS ratios that recover to their initial state at the end of the experiment (Figure 3d).

### 3.6. HL Effect on Pigment Composition

We surveyed the effect of HL treatment on the level of pigment concentration in *Synechocystis PSI*-*YFP*. The absorption spectra of intact cyanobacterial cells showed changes in the blue–green visible light region that indicated some HL-induced increase in carotenoid content (Figure 6a). A more detailed analysis of the chlorophyll content (calculated according to Ritchie (2006) [58]), showed an increasing trend in the control *Synechocystis PSI*-*YFP* cells while there was a stable trend in HL treated cells (Figure 6b). This is in line with the presence of slow growth of control sample that was rather absent in HL treated cells (Figure 1a). Interestingly, chlorophyll concentration per cell (Appendix A) slightly decreased during HL treatment, especially during the third (slow acclimation) phase. For that reason, we also analyzed the absorption spectra of the extracted pigment (Appendix A). These data were in line with the whole cell spectra (Figure 6a), showing an increase in carotenoids content, mostly during the third phase (120–360 min).

The HL-induced increase in carotenoids was analyzed by high-performance liquid chromatography (HPLC) (Figure 6c and Figure 7). The method identified the type of carotenoids present in the *Synechocystis PSI*-*YFP* cells (Figure 7a) together with their relative concentration changes during HL treatment (Figure 7b). The Chl/carotenoids ratio significantly decreased during HL treatment (in this case pointing to an increase in carotenoid content) and remained stable for the control sample (Figure 6c). This was in line with previous absorbance measurements (Figure 6a and Appendix A). The HPLC analysis additionally identified individual types of carotenoids that were either accumulated or remained unaffected during HL treatment; it includes β-carotene, zeaxanthin, and echinenone (Figure 7a). Moreover, we also noticed an additional peak at 32 min retention time, present only in the HL treated cells (Figure 7a). The peak was assigned as ɣ-carotene based on its spectrum, retention time and by comparing it with a commercial analytical standard (Appendix A). It showed that ɣ-carotene, an intermediate in the carotenoid biosynthetic pathway, accumulates during HL.

We further estimated kinetics of concentration changes in all carotenoids including this newly identified ɣ-carotene (Figure 7b). We observed almost no HL-induced changes in zeaxanthin, echinenone and β-carotene content (Figure 7b). This contrasts with a small increase in myxoxanthophyll during the final slow acclimation phase (120–360 min). The most significant increase, however, was in ɣ-carotene (Figure 7b). Ɣ-carotene increased significantly during the second phase (up to 30 min) and further increased during the third phase (Figure 7b). In total, the relative concentration of ɣ-carotene increased during HL from 0%–1% (typical for control samples) to 5% per total pigment (10% of total carotenoids). This suggests a new, yet unknown, role of ɣ-carotene accumulation during HL stress in cyanobacteria (for more details, see discussion).

## 4. Discussion

In this study, we observed how *Synechocystis PSI*-*YFP* copes during several hours of high-intensity light stress. We described kinetic changes of the cyanobacterial cells during HL treatment on several levels, starting from protein and pigment levels (e.g., carotenoid composition, see Figure 7), through sub-cellular and cellular level (changes in cell sizes, Figure 1b; in MDs, Figure 2), and to the whole cell suspension level (e.g., physiological parameters of photoprotection, i.e., NPQ and F_v_/F_m_ in Figure 4). We thus extended our previously published data that focused solely on PSI/PSII/and PBS co-localization in thylakoids [5]. We raised the hypothesis that after the initial fast response of cyanobacterial cells to HL (until 120 min), cyanobacterial cells tend to acclimate to the new light regime by a progressive return to a similar state as the initial physiological one of the no-stress condition. This is visible in several parameters including single-cell data (PSI/PSII/PBS ratios in Figure 3d) and bulk data of PSI/PSII ratio (Figure 5c). Therefore, in line with our previous single-cell data [5], we could define a three-phase response of *Synechocystis PSI*-*YFP* to acute HL stress: (1) fast response (0–30 min); (2) intermediate (30–120 min); (3) slow acclimation (120–360 min).

The overall cell physiology, concerning cell volume, responded to the acute HL stress very slowly, and it was visible only in the third slow acclimation phase (see 120–360 min in Figure 1). Our data showed (Figure 1b) that during the HL treatment, the ratio between bigger and smaller cells was significantly affected by HL. It might be an indication that *Synechocystis PSI*-*YFP* cells exposed to HL slow down their cell cycle, as the fraction of bigger cells was more abundant. The cell size control in cyanobacteria is rather complex [63]. It seems that sizes of cyanobacterial cells are not fully synchronized on the single-cell level because the cyanobacterial clock produces distinctly sized and timed subpopulations of cyanobacteria in both constant and light−dark conditions [63]. It does not contradict the fact that the parameters of the whole culture show synchronized behavior (see, e.g., bioreactor experiments [6,64]). Looking at our data we could hypothesize that, due to the HL, cells were “waiting” for the precise time to divide whilst coping with the light stress. This regulation of the timing of cell divisions could give an advantage in avoiding division during the energetically unfavorable periods. We can only speculate that the *Synechocystis*
*PSI*-*YFP* cell cycle might be delayed due to the energetically adverse time and the attempt to sustain the cell survival during the stress condition. Further studies will be necessary to confirm or disprove this.

We then stepped further from cell physiology to the sub-cellular level and explored TM organization of PPCs into MDs at the start and end of the experiment (Figure 2). We noticed that, although the fluorescence ratios of PPC emission of whole cells were similar between the initial and the final point of HL stress (Figure 3d), the spatial distributions inside the particular cells were altered (Figure 2c). These data are in line with the previous description of the TM mosaic in cyanobacteria [1,3,4,5]. The system of PSI/PSII/PBS organization into MDs changed significantly after 6 h of HL exposition; TM heterogeneity became less visible and all the three complexes were distributed more uniformly (compare Figure 2b,c, time 0 min and 360 min). The 6 h HL treatment decreased fluorescence in all three measured channels including PBS (Figure 3). This contrasts with the long-term acclimation (a shift during a week in light growth conditions, see [6]), when PBS fluorescence was the most stable, while PSI and PSII fluorescence were more equally distributed. Interestingly, the relative fluorescence emission of individual PPCs (normalized to sum of all three PPCs fluorescence) depicted the three-phase behavior that we noted before [5]. It shows that the three-phase response of the cyanobacterial thylakoids to 360 min of HL seems to be the typical strategy for *Synechocystis PSI*-*YFP* cells.

We compared the three-phase behavior detected at the single-cell level (Figure 3) with the analysis of the three photoprotective mechanisms: photoinhibition (F_v_/F_m_, Figure 4a); NPQ (Figure 4b) and PBS decoupling (Figure 4c,d—representing process of PBS detachment (energetical/physical) from PSII and/or PSI [18]). When PBS are energetically decoupled from photosystems, they cannot contribute to excessive excitation, protecting photosystems (see, e.g., [20]). The process can be measured as a relative increase in phycobilin fluorescence (Figure 4c). Indeed, we observed a significant increase in the PBS/PSII ratio at 90 min (during the intermediate phase) of the HL treatment (Figure 4d). These data indicate a temporal activation of PBS decoupling (around 90 min of HL) due to the acute light stress. However, our results did not show any sign of physical PBS decoupling from TM (Appendix A). Therefore, we suggest that the mechanism of PBS decoupling does not necessarily require a long-distance transport of PBS (e.g., from TM region into the center part of cell). Our data suggest that in some cases, only nanoscale re-arrangement changes near TM are necessary (Appendix A) to see signs of PBS decoupling in bulk measurements (Figure 4c,d). We proved that the process is triggered mostly during the second, intermediate phase.

In contrast to the PBS decoupling, the second photoprotective mechanism, blue light-induced NPQ, started to be activated already during the first phase of HL (0–30 min; Figure 4b). This proves NPQ to be the fastest response of the photosynthetic membrane to excess light [65]. Our data showed a continuous increase in NPQ during the fast response and intermediate phases (up to 120 min; Figure 4b). During the third (slow acclimation) phase the absolute value of NPQ did not change; however, NPQ becomes more flexible as it is visible in the fast NPQ recovery (Figure 4b). In contrast to NPQ activation, which is triggered by OCP [66], the recovery process requires another protein—fluorescence recovery protein (FRP) [67]. The data suggest that the amount or activity of the proteins could increase during the third—slow acclimation—phase. This is just a hypothesis that will need to be confirmed/disproved by direct measurements.

The other photoprotective mechanism we studied during acute HL stress was photoinhibition. We showed a clear sign of photoinhibition during HL stress based on both fluorescence parameters (Figure 4a) and biochemistry (Figure 5b). A faster response was visible in F_v_/F_m_ that decreased already at 10 min (fast phase in Figure 4a). However, the actual PSII degradation (visible as PSII monomerization, Appendix A) was slower and appeared later during the intermediate phase (around 60 min of HL; Figure 5b and Appendix A). These biochemical data were in line with previous observations (see, e.g., [68]); however, we have newly proved that the actual process of PSII monomerization is delayed after the first signs of photoinhibition, visible based on fluorescence.

The biochemical data showed that acute HL stress was linked with faster turnover of pigmented proteins of PSII (see, e.g., Figure 5). These protein complexes contain tens of pigments (chlorophyll, carotenoids) whose concentrations could be affected by protein degradation. Therefore, we have also estimated HL-induced changes in pigment composition by several methods (Figure 6 and Figure 7). It is known that carotenoids in phototrophs have an essential role in photoprotection [31,32] and they are also involved in regulation of biological membranes (e.g., fluidity, see [69]) and pigment–proteins photochemistry (e.g., light harvesting efficiency), where they act as allosteric regulators (see, e.g., [47,48]). In cyanobacteria, the most typical carotenoid involved in photoprotection is echinenone, which is present in OCP protein involved in blue light-induced NPQ in PBS [67]. However, we did not detect any change in its concentration during 6 h of HL (Figure 7).

We found that the carotenoid content responds to acute HL stress in a surprisingly fast manner. For instance, ɣ-carotene started to accumulate significantly already during the second phase (60–120 min; Figure 7b). Subsequently, during the slowest acclimation phase (120–360 min) myxoxanthophyll content is also increased (Figure 7b). Interestingly, the carotenoid accumulation in the intermediate phase correlates with significant changes in PSII, specifically with the appearance of PSII monomers (Figure 5b and Appendix A). We can tentatively suggest that higher myxoxanthophyll and ɣ-carotene contents can facilitate D1 protein synthesis and its assembly into PSII during accelerated photoinhibition (Figure 4a). It has been already suggested that proteins are mobilized during HL stress [15] or when they are disassembled from larger complexes [70]. Indeed, carotenoids (echinenone, zeaxanthin, myxoxanthophyll) were suggested to be involved in the PSII repair mechanism [71]. We thus hypothesize that the increased level of carotenoids (relative to chlorophylls, see Figure 6c) facilitates protein complexes’ disassembly (especially PSII) and their movement into assembly centers, regarding which localization is still intensively discussed [13,72]. In fact, the structural importance of carotenoids for assembly of protein complexes, in this case PSI trimer, has been already proved [73].

Apart from the well-known photoprotective role of carotenoids (either direct photoprotection or in a tuning of protein functions [47,48]), they also have a clear structural effect in thylakoids [69]. Free carotenes and xanthophylls seem to be present in the hydrophobic region of the membrane bilayers. It has been hypothesized that they can affect membrane fluidizing effects [74]. For instance, myxoxanthophyll seems to be important for the TM structure [50,75]. Therefore, the increased myxoxanthophyll content we detected could affect membrane fluidity. The preferential synthesis of myxoxanthophylls in HL has already been observed [76], and they could possibly make the membrane more fluid, helping the photosynthetic apparatus during PSII disassembly/assembly in photoinhibition. Indeed, the “proteins mobilization” during HL stress has already been suggested for cyanobacteria [15]. For those reasons, we hypothesize that carotenoids, in addition to photoprotection, could play a role in modulation of membrane fluidity, important for protein mobility [77]. This hypothesis, however, needs to be tested by specific mutants without carotenoids.

The progressive ɣ-carotene accumulation on HL (Figure 7b) was the most surprising effect we observed. This carotenoid represents the last common precursor to the biosynthesis of all common carotenoids. Its HL-induced increase has not been reported in *Synechocystis* sp. PCC 6803 cells, yet ɣ-carotene was not usually included in HPLC pigment analyses, as its concentration (at non-stress condition) is often almost too insignificant to be detected (less than 1% of all carotenoids). However, we undoubtedly proved a small, but reproducible ɣ-carotene accumulation in HL, up to almost 10% of all carotenoids. Therefore, it does not seem to be caused by some malfunction of the carotenoids’ biosynthesis pathway. Moreover, it is possible that ɣ-carotene increase affects also myxoxanthophyll accumulation. The monocyclic ɣ -carotene represents a branching point towards either β-carotene or myxoxanthophyll [78] that could possibly act as two competing routes during HL stress. The increased level of ɣ-carotene could thus represent a physiological response of cyanobacterial cell physiology during acute HL stress. One might suggest that its accumulation brings an advantage in that it provides a higher carotenoid pool during HL, a benefit for photoprotection (e.g., against ROS). Interestingly, the accumulation of carotenoid intermediates, (e.g., intermediates of the xanthophylls cycle) as an “economic” mechanism of photoprotection, was shown in eukaryotic phototrophs including the diatom *Phaeodactylum tricornutum* [79] and in the prasinophycean alga *Mantoniella squamata* [80]. In these organisms, the pathway intermediates were transformed into light harvesting and photoprotective pigments upon return to low light illumination, reducing the costs of a de novo synthesis of light-harvesting xanthophylls. Therefore, the increased pool of ɣ-carotene during HL (Figure 7b) in cyanobacteria could act in a similar way. The evaluation of the ɣ-carotene for photoprotection remains a task for future experiments.

## 5. Conclusions

We described a three-phase response of the *Synechocystis PSI*-*YFP* cell to acute HL stress, first noted by Konert et al. [5]. We surveyed the HL stress response of cyanobacterial cells at all possible levels, starting from the protein/pigment level, before analyzing the cellular level and finally the physiological parameters of photoprotection. The first phase, a “fast response phase“ (0–30 min), showed a typical phenomenon of photoinhibition (Figure 4a and Figure 5) and blue light-induced photoprotective non-photochemical quenching (NPQ) localized in PBS (Figure 4b). Later on, during the second phase (30–120 min)—“intermediate phase”—we detected accumulation of ɣ-carotene that correlated with depletion in PSII dimer and with functional decoupling of PBS from photosystems. The carotenoid accumulation was further stimulated in the last “acclimation phase” (120–360 min) when fluorescence properties of PPCs per cells progressively returned to their initial values (Figure 3d). It indicated transitions of cells into a new physiological equilibrium more suitable for the acute HL stress conditions. The new HL-acclimated state was identifiable by increased content of carotenoids (ɣ-carotene and myxoxanthophyll, Figure 7b) and by more even spatial distribution of PSI, PSII and PBS between microdomains (Figure 2b,c). We suggest that carotenoid accumulation might be involved in the maintenance of the new membrane and protein structure at HL condition. It is unclear whether the transition into the new steady state after 6 h of HL also requires a delay in the cyanobacterial cell cycle, as this was indicated by HL-induced changes in the cell sizes during HL treatment (Figure 1b). This hypothesis, however, requires additional experiments to be verified.

## Figures and Tables

**Figure 1 cells-10-01916-f001:**
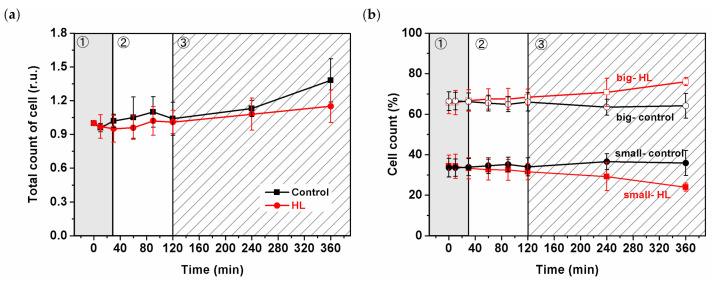
Changes in the number of *Synechocystis PSI*-*YFP* cells measured by cell counter during high-light (HL) treatment. (**a**) Relative changes in total cell count obtained independently of cell sizes. Data are normalized to time 0 min. (**b**) Relative changes in cell count of small (diameter ≤ 1.85 µm) and big (diameter >1.85 µm) cells. Black lines = control; Red lines = HL treatment. The three phases of cyanobacterial cell response to HL (see Konert et al. 2019 [5]) are marked by different background colors and numbers (grey—fast response; white—intermediate phase; striped—acclimation phase). Data represent averages and SD from three biological replicates, the starting culture (t = 0 min) typically contains *n* = 10^6^ cells per mL.

**Figure 2 cells-10-01916-f002:**
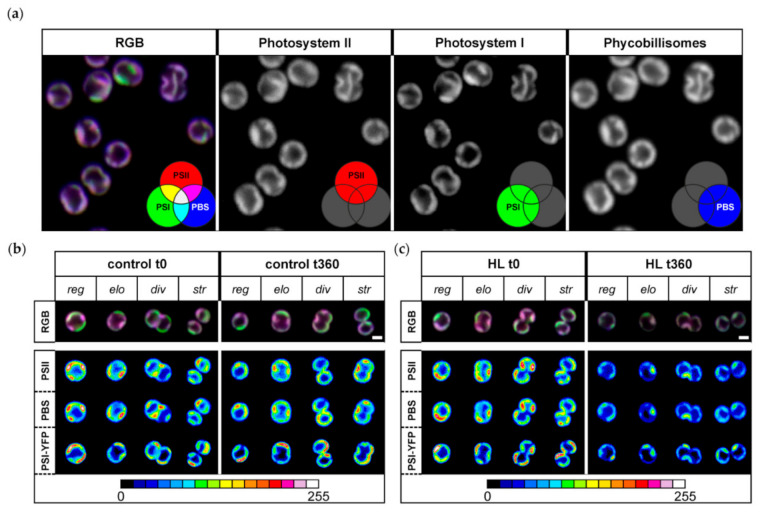
Microdomain (MD) organization of thylakoid membrane in *Synechocystis PSI*-*YFP* during HL treatment. (**a**) Typical RGB picture of *Synechocystis PSI*-*YFP* cells obtained by three channel confocal imaging (PSII channel—red; PSI–YFP channel—green; PBS channel—blue). Colors reflect PSI/PSII/PBS co-localization and the most dominant microdomains, namely magenta MD (dominant PBS and PSII, less PSI-YFP) and green (dominant PSI-YFP, less PSII and PBS) as described by Strašková et al. 2019. Panels (**b**) and (**c**) represent change in MD organization in control (**b**) and HL treated cells (**c**) at the beginning (t = 0 min) and the end of the experiment (t = 360 min). The four typical cell shapes are presented independently (reg = regular, elo = elongated, div = dividing, str = string). The first row shows three channels’ pictures (RGB, 24-bit) with PSII, PBS, and PSI-YFP co-localization. The second, the third, and the fourth rows depict intensity of single-channel fluorescence of PSII, PBS, and PSI-YFP, respectively. Colors reflect intensity of fluorescence signal per channel (heatmap images) in the 8-bit scale 0–255 (see the color scale bar).

**Figure 3 cells-10-01916-f003:**
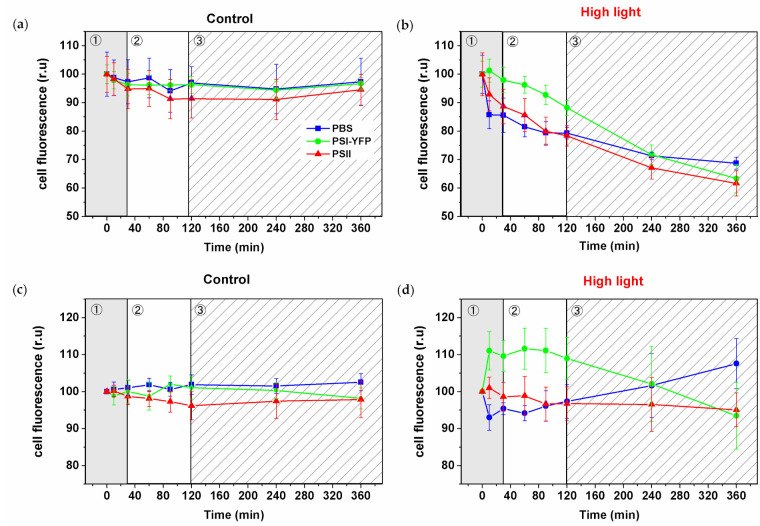
High-light induced changes in PSI-YFP, PSII, and PBS fluorescence emission detected from the single cells of *Synechocystis PSI*-*YFP*. Single-cell fluorescence intensities were calculated from three channels’ confocal images (see, e.g., Figure 2) acquired at particular time spots during HL treatments. Panels (**a**) and (**b**) represent normalized fluorescence changes of single-cell PSII, PSI-YFP, and PBS emission for control (**a**) and HL treated sample (**b**). Data were normalized to fluorescence intensity at t = 0 min for every channel independently. Panels (**c**) and (**d**) show relative fluorescence changes in PSI-YFP, PSII, and PBS when the presented values are normalized to the sum of all three PPCs’ emission (PSI + PSII + PBS) and to the initial value at t = 0 min for each channel individually. Data represent control (**c**) and HL treated cells (**d**) of six independent biological replications. The total count of cells used for analysis was *n* = 5844. The three phases of cyanobacterial cell response to HL (see Konert et al. (2019) [5]) are marked by different background colors and numbers (grey—fast response; white—intermediate phase; striped—acclimation phase).

**Figure 4 cells-10-01916-f004:**
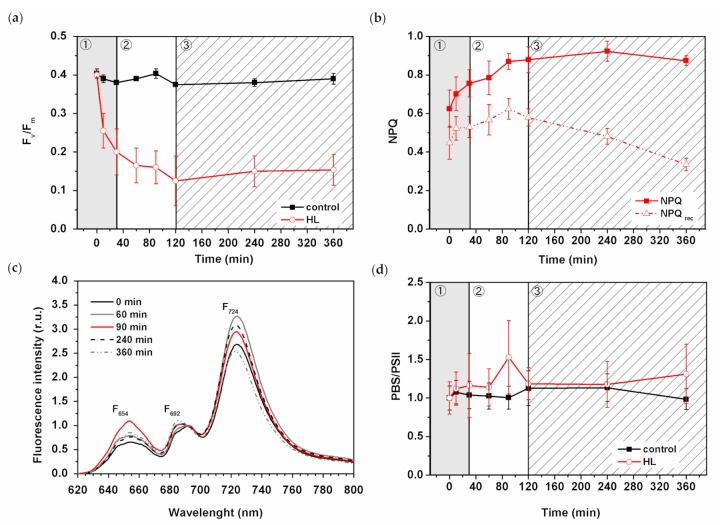
Effect of HL stress on photoprotective mechanism in *Synechocystis PSI-YFP* cells. (**a**) Change in maximal quantum yield of PSII (Fv/Fm) due to photoinhibition in cell suspension. Black/red lines represent control/HL treated sample; *n* = 3. (**b**) Extent of blue light-induced non-photochemical quenching on light (NPQ—squares) and its recovery in dark (NPQrec—triangles) during HL stress; *n* = 4. (**c**) Typical curve of low temperature (77K) fluorescence emission spectra (excitation 530 nm) for cells exposed to HL for different times (0 min—straight black line; 60 min—straight grey line; 90 min—red line; 240 min—dashed black line; 360 min—dashed grey line). F654, F724, F724 represent the positions of the main emission peaks of PBS, PSII and PSI, respectively. (**d**) The PBS/PSII ratios estimated as F654/F692 from panel (**c**) for both HL and control cells; data were normalized to values at t = 0 min; *n* = 4. The three phases of cyanobacterial cell response to HL (see Konert et al. (2019) [5]) are marked by different background colors and numbers (grey—fast response; white—intermediate phase; striped—acclimation phase). All data in panels (**a**,**b**,**d**) represent averages and standard deviations.

**Figure 5 cells-10-01916-f005:**
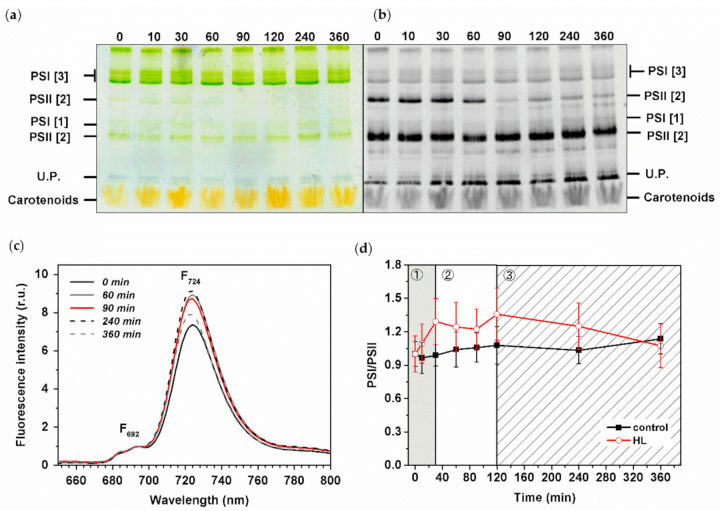
Biochemical and spectroscopic analysis of pigment–protein complexes of HL treated *Synechocystis PSI*-*YFP cells*. Thylakoid membranes were isolated from the cells and analyzed by CN-PAGE. Gels were imaged as a color-scanned picture (**a**) or as chlorophyll autofluorescence pictures (**b**). Numbers represent time points (in minutes) of HL treatment. Designation of complexes: PSI [3], PSI [1] trimeric and monomeric PSI complexes; PSII [2] and PSII [1] dimeric and monomeric PSII core complex; U.P. = unbound proteins. Each loaded sample contained 5 µg of Chl. (**c**) Normalized 77K fluorescence emission spectra at 692 nm (450 nm excitation) at different times of HL treatment. Straight black line = time 0 min; straight grey line = time 60 min; red line = time 90 min; dashed black line = time 240 min; dashed grey line = time 360 min. Positions of F_692_ and F_724_ representing PSII and PSI maximal emission, respectively, are marked. (**d**) Spectroscopic analyses of PSI/PSII ratio based on 77K fluorescence (excitation at 450 nm) for HL and control cells. PSI/PSII ratio deduced from the intensity of fluorescence of F_692_ (PSII maxim) and F_724_ (PSI maxim). Data represent averages and SD of four biological replicates and normalized to values at 0 min. The three phases of cyanobacterial cell response to HL (see Konert et al. (2019) [5]) are marked by different background colors and numbers (grey—fast response; white—intermediate phase; striped—acclimation phase).

**Figure 6 cells-10-01916-f006:**
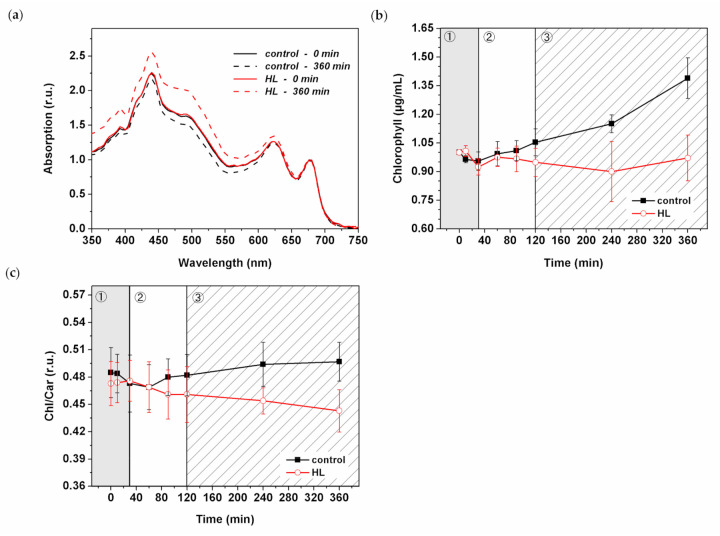
Spectroscopic and HPLC analysis of pigment composition during HL treatment of *Synechocystis*
*PSI*-*YFP* cells. (**a**) Absorption spectra. Data were normalized to the peak corresponding to PSII (664–665 nm). Straight black line = control time 0 min; dashed black line = control 360 min. Straight red line = HL time 0 min; dashed red line = HL time 360 min. Average of four independent biological measurements. (**b**) Chlorophyll concentration was estimated spectroscopically according to Ritchie et al. (2006). Black line = control; red line = HL. Data represent average of four independent biological measurements that are normalized to time 0 min. (**c**) Chlorophyll/total carotenoids ratio was estimated based on HPLC analysis. All detected carotenoids were taken into consideration, *n* = 4. Black line = control; red line = HL. Presented HPLC data represent averages and SDs of four independent biological measurements. The three phases of cyanobacterial cell response to HL (see Konert et al. (2019) [5]) are marked by different background colors and numbers (grey—fast response; white—intermediate phase; striped—acclimation phase).

**Figure 7 cells-10-01916-f007:**
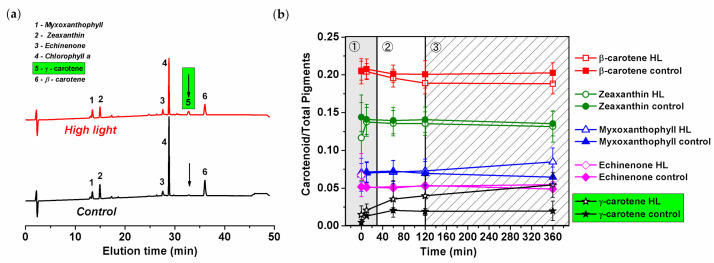
HPLC chromatogram and kinetic changes in carotenoid composition in *Synechocystis PSI*-*YFP* in HL and control culture. (**a**) HPLC chromatograms of the control and HL treated samples for 360 min. Position of the peaks was identified based on literature (see description). The peak presented only in HL cell (no. 5, see arrows, green bar), was identified by comparison of its retention time and absorbance with the analytical standard as ɣ-carotene (see Appendix A). (**b**) Kinetics of carotenoid accumulation during HL treatment: control sample—closed symbols; and HL—open symbols. Data represent averages and SD obtained from three biological replicates. The three phases of cyanobacterial cell response to HL (see Konert et al. (2019) [5]) are marked by different background colors and numbers (grey—fast response; white—intermediate phase; striped—acclimation phase).

## Data Availability

Not applicable.

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
