# Peer review of "Gradual Response of Cyanobacterial Thylakoids to Acute High-Light Stress—Importance of Carotenoid Accumulation"

_cells, 2021, doi:10.3390/cells10081916_

Round 1

Reviewer 1 Report

The authors studied in detail how a cyanobacterium Synechocystis 6803 mutant, in which photosystem I was fused with GFP, responded to high light treatment in a three-phase process. The study applied multiple techniques including confocal imaging, fluorescence/absorption spectroscopy, pigment analysis and protein electrophoresis. While the study showed some interesting results, it is a descriptive work than exploring the mechanisms underlying the HL acclimation. I have the following concerns and suggestions:

  1. The results of this study were obtained from a PSI-YFP mutant. There is no data or specification on the differences in cell growth and physiology between the WT and PSI-YFP mutant under HL treatment. A comparison of the two strains should be provided.
  2. Line 54-56: There have been several research articles and reviews on the heterogeneity of cyanobacterial thylakoid membranes. The study should acknowledge the previous studies (PNAS 2008, 105:4050; Plant Physiol 1994, 106:251; Plant Physiol 2012, 158:1600; Nat Commun 2021, 12:3475; Nat Plants 2020, 6:869; Annu Rev Microbiol 2020, 74:633), in addition to discussing their own work (BBA 2019).
  3. Synechocystis thylakoid membranes have irregular structures. It is unclear if the so-called microdomains were an implication of irregular shapes of thylakoid membranes or clusters of photosynthetic complexes.
  4. Does the reduced abundance of photosynthetic proteins reflect a decrease in thylakoid membrane layers or diluted membrane distribution of proteins?
  5. How the three phases of the acclimation responses were identified was not explained in detail, since many of the changes were not hugely significant.
  6. Figure 1b: Does 1.6 micrometer represent the diameter of cells in all different directions? Why was it used as a threshold to categorize the big and small cells? What are the average sizes of big and small cells? It is better to specify the difference in size using the percentage. In addition to the size, the cells differ also in types (regular, elongated, dividing, or string, line 174-175). Were they considered and separated in Figure 1b?
  7. Line 305-312: I do not fully understand why cell division was slightly affected but no major differences were found in cell types. Bigger cells do not represent simply the cells that had interrupted division.
  8. Line 358-359: what is the analytical data to support a more even distribution after 6 h of HL?
  9. Figure 5: The results showed in cells there are more PSII monomers than PSII dimers. Is it true and consistent with other results? Does GFP tagging affect PSI assembly?
  10. Figure 6: the decrease in Chl content is not significant based on the results. T-tests should be conducted. Nevertheless, the trend of the changes in Chl content is not consistent with the confocal results that showed a reduction in Chl content (see Fig. 3b).

Other comments:

  • It is a long article, which should be revised and reformatted. For instance, the introduction is too long and discussed several open questions. It is recommended that the authors focus on the questions that have been addressed in this work and delete those irrelevant descriptions. Moreover, Discussion has some overlaps with the Results. It can be more focused on the main findings and biological context. The Conclusions session is also very long, including a summary at the end. This can be largely simplified. “nanoscale level of pigments and proteins, to microscale level of membrane and whole cell” has not been determined in this study and sounds overinterpreted.
  • Abstract mentioned three photoprotective mechanisms, whereas four mechanisms were stated in Introduction but only three were described in more detail. It is essential to keep consistency.
  • There are many language errors, which I cannot list all here. Some examples:

Line 39: the cytoplasm
Line 40: light-dependent photosynthetic reactions
Line 44: the TM surface
Line 47: on the molecular level
Line 91: delete “in addition to photoinhibition,”. The second …

  • Line 122-125: it is relevant to discuss the recent paper on should be discussed here.
  • Line 132-136: the recent paper on cyanobacterial thylakoid membrane biogenesis induced by HL should be discussed (Nat Commun 2021, 12:3475).
  • Line 330-331: PBS cannot be inside of thylakoids
  • Figure 1: it is important to specify how many cells were counted and do t-tests for differences, better than r.u.
  • it is unclear how fluorescence values can be normalized to the sum of 3 PPCs. Fluorescence of the 3 PPCs was detected at different wavelengths, for which the microscope detector has variable sensitivities. Thus, the detected fluorescence intensities cannot be comparable. Please specify the data analysis process.
  • please specify the numbers of cells analyzed.
  • Were the results achieved from multiple biological replicates? If so, please add error bars and specify the number of biological replicates.

Author Response

We are thankful for the time and effort that the reviewer put into reading commenting our manuscript.  We hope, that our manuscript is now more readable and improved due to the implementation of all these comments and objections.  Please find enclosed our response (point by point) in the attached file

Reviewer 2 Report

The manuscript “Gradual response of cyanobacterual thylakoids to acute high-light stress-importance of carotenoids accumulation” is a nicely done study on the responses and mechanisms of acclimation to high light (HL) in Synechocystis sp. PCC 6803. The authors employ with expertise a complete set of biophysical and biochemical techniques yielding clear and convincing results. I have just some minor comments that could be used to clarify some aspects of the, otherwise excellent, manuscript:

-Some sentences and paragraphs are unclear and should be rewritten (e.g. lines 143-150; 276-278; 635-636).

-A significant part of the discussion is just a repetition of the results section, these redundancies could be removed to shorten the text and to gain flow.

-Authors conclude that one of the more relevant findings of the study is the rapid accumulation of gamma-carotene in response to HL. Then the role of this carotenoid should be more precisely described in the context of current knowledge on carotenoid biosynthesis in cyanobacteria. At this regard, one possibility is that gamma-carotene is the precursor of myxoxanthophyll, and this could be the reason why these two carotenoids increased in response to HL. Monocyclic gamma-carotene could then represent a branching point towards beta-carotene or myxoxanthophyll with two competing routes. Authors suggest that gamma-carotene might facilitate D1 synthesis and incorporation into PSII (lines 728-729), but a mechanistic support is missing.

Author Response

We are thankful for the comments, we tried to implement all of them. Please find enclosed file with our pint-to-point response.

Round 2

Reviewer 1 Report

I am happy with the changes made and extra results provided by the authors in the revised manuscript.